# FEABench: Evaluating Language Models on Real World Physics Reasoning Ability

**Nayantara Mudur**[1,2*]    **Hao Cui**[1]    **Subhashini Venugopalan**[1]

**Paul Raccuglia**[1]    **Michael Brenner**[1,2]

**Peter Norgaard**[1]
[1]Google Research    [2]Harvard University
{nmudur,jacksoncui,vsubhashini,praccu,mbrenner,pnorgaard}@google.com
nmudur@g.harvard.edu

## Abstract

Building precise simulations of the real world and using numerical methods to solve quantitative problems is an essential task in engineering and physics. We present FEABench, a benchmark to evaluate the ability of language models (LLMs) and LLM agents to simulate and solve physics, mathematics and engineering problems using finite element analysis (FEA) software. We introduce a multipronged evaluation scheme to investigate the ability of LLMs to solve these problems using COMSOL Multiphysics®. We further design an LLM agent equipped with the ability to interact with the software through its Application Programming Interface (API), examine its outputs and use tools to improve its solution over several iterations. Our best performing strategy generates executable API calls 88% of the time. However, this benchmark still proves to be challenging enough that the LLMs and agents we tested were not able to completely and correctly solve any problem. LLMs that can successfully interact with and operate FEA software to solve problems such as those in our benchmark would push the frontiers of automation in engineering. Acquiring this capability would augment LLMs' reasoning skills with the precision of numerical solvers and advance the development of autonomous systems that can tackle complex problems in the real world.

## 1 Introduction

Several works have demonstrated the significant potential of large language models (LLMs) on analytical mathematical and scientific reasoning [1, 2, 3, 4, 5, 6], and on programming tasks in general-purpose languages like Python [7, 8, 9]. However, many quantitative tasks that are the cornerstone of engineering and scientific workflows require numerical analysis performed with sophisticated computational modeling software. For example, the development of a modern smartphone requires detailed modeling of the mechanical, thermal, and electrical behaviors of its many subcomponents. Finite element analysis (FEA) [10] is the approach typically used. This involves approximating the partial differential equations that describe the physical behavior of a system by building discretizations (or meshes) over geometries. This is then solved until convergence using numerical solvers. The vast relevance of FEA to domains like mechanical, biomedical and aerospace engineering, optics, and fluid dynamics has given rise to software such as Ansys®[11], Abaqus® FEA [12], and COMSOL

---

*Work mainly done as a student researcher at Google Research.

38th Conference on Neural Information Processing Systems (NeurIPS 2024).

Figure 1: *Left:* A problem description. *Right:* Problems by domain.

Table 1: Summary of Evaluation Metrics

| METRIC | ARTIFACTS | SKILLS MEASURED | | |
| --- | --- | --- | --- | --- |
| | | Correctness | Alignment | Physics Reasoning |
| Executability | API Messages | ✓ | | |
| Model Tree Score | Model Tree | | ✓ | |
| Code Diff Score | Code | | ✓ | |
| Physics Metrics | Physics Code | | | |
|    Interface Factuality | | ✓ | | |
|    Recall Metrics | | | ✓ | ✓ |
|    Feature Dimension | | ✓ | | ✓ |
| Target Value Metrics | Output Table | ✓ | ✓ | ✓ |

Multiphysics®[13, 14], that are indispensable to modelling complex systems with the interplay of non-trivial geometries, and multiple physical phenomena.

In this paper, we begin to bridge this gap by examining the ability of LLMs to solve problems using finite element analysis (FEA) by generating Java API calls. This task requires reasoning over the problem and the ability to successfully operate the software. We selected COMSOL Multiphysics®because it supports a wide range of physics models and can be applied to applied mathematics problems involving optimization or differential equations, to problems involving complex geometries. Because the FEA workflow is relatively canonical, the reasoning approach for modeling is similar to other FEA software, and all problems typically share a conceptual breakdown into a sequence of blocks that involve defining (1) Geometry, (2) Material properties, (3) Physics, (4) Meshing (5) Numerical Analysis and Solver settings, and (6) Postprocessing (details in Appendix C.2). COMSOL Multiphysics®Version 6.1 is used in this work.

Our contributions are the following: We create a benchmark of 15 quantitatively verifiable problems. These problems require numerical methods to solve, and the target values are expected to be largely independent of the modeling software. We measure different state-of-the-art LLMs on their performance with a multi-score evaluation strategy. Finally, we design a multi-turn LLM agent that interacts with the API and improves upon its answers over several iterations and examine its performance.

## 2 Dataset

The benchmark problems are derived from tutorials in the COMSOL Multiphysics®Application Gallery [15] which are often based on established validation problems or other sources [16, 17]. The input is a problem description describing a problem with a specific target quantity that needs to be computed (Figure 1). The problems span a range of real world / mathematical systems including the dynamics of a Lorenz attractor, heat transfer, and a Bragg reflector. Each entry consists of the following main fields: (1) **Model Specifications:** A condensed yet fully-specified description of the problem, and the physics / boundary conditions. This description is unambiguous in terms of details such as material properties or dimensions. (2) **Selection Information:** An engineer would rely on visual information from the Graphical User Interface (GUI) (Figure 3) for spatial information,

particularly the numerical representation of geometrical entities, like points or boundaries. Since the agents do not have visual access to the GUI, we augment the model specifications with this information, to supply the model with the necessary representation of spatial information in the GUI. This information will be valid as long as the agent chooses to construct the geometry in a manner reasonably similar to how the ground truth (GT) geometry was created. (3) **Plan:** An explicit step-by-step description in natural language of the steps to be followed to solve the problem using COMSOL Multiphysics®. Either of the following comprise self-sufficient tasks for the agent to solve: a) Model Specifications + Selection Information (designated **ModelSpecs**), or b) Plan (**Plan**). (4) **Target Description:** A brief phrase describing the quantity that needs to be computed. (5) **Target Value:** The correct value of the target physical quantity. Note, the target values are expected to be largely independent of the choice of modeling software. (6) **Ground Truth Code:** Ground truth code, that if executed, is able to successfully solve the problem and export the target value to a file. (7) **Model Tree:** Executing COMSOL Multiphysics®calls essentially build a model that can be regarded as modifying a tree with certain predefined 'branches' such as 'geometry' and 'physics'. The model generated by executing code can thus be represented in a condensed form as a model tree (see Appendix A.3 for an example). Converting a tutorial to a benchmark problem requires ensuring that a verifiable artifact (the target) can be derived from it, generating inputs and ensuring they have all the information needed to solve the problem, generating the GT solution and verifying that it exports the correct target value (Appendix A).

# 3 Evaluation Metrics

Our evaluation metrics measure the correctness of the solution, even when a target value could not be exported, (Table 1). These additional metrics offer the advantage of being continuous, unlike the relative error, which can only be computed if the solution exported a valid target value. Metrics denoted by † require execution. We delineate the metrics, and the facets they probe here: (1) **Executability†:** 'Correct' (executable) lines as a fraction of parsed API calls in an LLM solution. (2) **Model Tree Score†:** Similarity score between the LLM code's model tree and the GT tree. This normalized so that a solution with no lines of code is scored 0. If equivalent to the GT code, it would be 1. This measures the *alignment* of the model's solution path with a successful path. (3) **Code Similarity Score:** Similarity score between the solution and the GT code. While this also measures alignment with the GT solution, two different code blocks could generate equivalent model subtrees. We mainly report this metric for completeness, and to motivate the need for the domain-specific metrics we introduce here. (4) **Physics Metrics:** The code is a basis to represent the actions the LM takes to model the problem. Since the physics block is the most challenging (Figure 2), we additionally analyze the LLM's physics action sequence by parsing and evaluating the physics commands. The most basic physics action sequence involves: Create Interface (eg: HeatTransfer) → Create Feature under Interface (eg: TemperatureBoundary) → Modify Feature Properties (eg: T0, to set a temperature). Our Physics Metrics include (a) *Interface Factuality*: How many defined interfaces were *not* hallucinated? (b) *Interface / Feature / Feature Property Recall*: How many interfaces / features / feature properties created / modified by the GT solution were also in the LLM solution? (c) *Feature Dimension:* For features created by both, does the feature's spatial dimension match? Some nested physics metrics, such as 'Feature Dimension' might not be valid for a specific problem, in case there was no matching feature between the ground truth and the LLM code: we mask out these problems while computing the means for that specific metric. (5) **Target Relative Error†**: There are various ways in which exporting the correct value may fail: a) the code fails to export b) it exports an 'empty' table or, c) it exports an incorrect table, such as a default value or the wrong quantity. We ask an LLM to check that the exported table matches the target description and that the exported quantity is not a default or a value reproduced as is from the problem description, and to parse the response, if so. **Valid Target** is the number of problems in the benchmark for which the LLM successfully judges the exported table to be valid. We then compute the relative error between the last value in the exported table and the GT answer. **Relative Error | Strict** computes the mean relative error only over problems for which Valid Export is True, AND the relative error is less than 10%. Relative Error | Strict is our key metric for whether the LLM truly solved a problem.

Table 2: Code Metrics: Comparison on **ModelSpecs** across LLMs.

| Experiment | Executability | Model Tree Score | Code Similarity | Valid Target |
|---|---|---|---|---|
| Claude 3.5 Sonnet [18] | **0.79**±0.03 | **0.69**±0.07 | **0.19**±0.03 | **1**/15 |
| GPT-4o [19] | 0.78±0.03 | 0.56±0.06 | 0.17±0.03 | 0/15 |
| Gemini-1.5-Pro [20] | 0.60±0.05 | 0.46±0.07 | 0.17±0.03 | 0/15 |

Table 3: Physics Metrics: Comparison on **ModelSpecs** across LLMs.

| Experiment | Interface Factuality | Interface Recall | Feature Recall | Feature Property Recall | Feature Dimension |
|---|---|---|---|---|---|
| Claude 3.5 Sonnet | **0.85**±0.10 | **0.71**±0.13 | **0.80**±0.10 | **0.22**±0.10 | **0.95**±0.05 |
| GPT-4o | 0.79±0.11 | 0.64±0.13 | 0.55±0.12 | **0.22**±0.11 | **0.95**±0.05 |
| Gemini-1.5-Pro | 0.54±0.14 | 0.43±0.14 | 0.39±0.10 | 0.15±0.09 | 0.86±0.14 |

Table 4: Code Metrics: Comparison across tasks, prompts and agents.

| Experiment | Executability | Model Tree Score | Code Similarity | Valid Target |
|---|---|---|---|---|
| **ModelSpecs** : One-Shot | 0.60±0.05 | 0.46±0.07 | 0.17±0.03 | 0/15 |
| **ModelSpecs** : PhyDoc In-Context | 0.62±0.05 | 0.58±0.07 | 0.15±0.02 | 1/15 |
| **ModelSpecs** : Multi-Turn Agent | **0.88**±0.03 | 0.56±0.08 | 0.17±0.03 | **2**/15 |
| **Plan** : One-Shot | 0.54±0.03 | 0.39±0.03 | **0.21**±0.03 | 0/15 |
| **Plan** : PhyDoc In-Context | 0.59±0.05 | **0.59**±0.06 | 0.20±0.02 | 0/15 |

Table 5: Physics Metrics: Comparison across tasks, prompts and agents.

| Experiment | Interface Factuality | Interface Recall | Feature Recall | Feature Property Recall | Feature Dimension |
|---|---|---|---|---|---|
| **ModelSpecs** : One-Shot | 0.54±0.14 | 0.43±0.14 | 0.39±0.10 | 0.15±0.09 | 0.86±0.14 |
| **ModelSpecs** : PhyDoc In-Context | **1.00**±0.00 | 0.71±0.13 | 0.48±0.10 | 0.08±0.07 | 0.59±0.16 |
| **ModelSpecs** : Multi-Turn Agent | 0.93±0.07 | **0.79**±0.11 | **0.75**±0.09 | 0.24±0.10 | 0.89±0.07 |
| **Plan** : One-Shot | 0.38±0.14 | 0.36±0.13 | 0.43±0.11 | **0.32**±0.11 | 0.79±0.15 |
| **Plan** : PhyDoc In-Context | 0.85±0.10 | 0.57±0.14 | 0.47±0.11 | 0.13±0.07 | **0.93**±0.07 |

## 4   Results

**Comparison across LLMs**   Our baseline experiment queries different frontier LLMs with a one-shot prompt (see Appendix G). Table 2 records the means and standard errors on the means across problems. Mean executability is in the range 0.60-0.79, implying that the LLMs are familiar with the higher-level grammar and syntax of COMSOL Multiphysics®API code. However, they are prone to hallucinating the choice of interface (factuality between $[0.54 - 0.85]$). This is likely a significant contributor to the non-executable lines because an invalid interface renders all physics lines of code under this interface invalid. Claude 3.5-Sonnet has the best performance in the baseline setting. See Appendix F for a qualitative analysis.

**Comparison across Tasks and Agent Settings**   We fix the LLM to Gemini-1.5-Pro, and compare performance with the list of physics interfaces and features (PhyDoc In-Context) added to the prompt and in an interactive multi-turn agent setting. The agent is equipped with an 'evaluator' that both executes the code and records API replies and when executability crosses 90%, examines the code and the output table, and returns feedback, in an 'LLM-as-Judge' fashion. It thus computes a 'fitness' for each solution *without* having access to the GT target value. The fitness is used to track the best solutions. The CONTROLLERAGENT calls a CORRECTORSUBAGENT that calls tools that aid it in correcting the code given the 'current' code and feedback, execution history and the result of tool calls. An initial set of 20 samples are generated using PhyDoc In-Context and the best solutions are corrected for 20 steps (see Appendix C.2 for details). The best of all

solutions is then identified and evaluated. Both PhyDoc In-Context and the agentic setting significantly reduce interface hallucinations, and the agent's mean executability is 0.88 (vs 0.62). *Thus, grounding the LLM with information about or interaction with the API boosts performance.* This indicates that at least some of the difficulty the LLMs encounter is due to their lack of knowledge of valid arguments to the API commands. We further examine whether the **Plan** task is easier. The comparison between task versions is of interest since both demand slightly different skills.

For a person attempting to solve this task, **ModelSpecs** requires the individual to both *infer* implicit engineering and physical reasoning decisions to be made (e.g.: the LLM needs to infer that the correct representation of a cylinder's 2D cross-section is a rectangle) and further translate this to valid API calls. **Plan** explicitly describes all steps to be followed in natural language and requires the LLM to only *translate* the steps describing interactions with the GUI to valid calls. The comparison between the two tasks offers one way to decouple the complexity of making correct modelling decisions from the difficulty of translating decisions into syntactically correct API calls. If an LLM fared poorly at making the right modelling decisions but could reliably translate natural language instructions to API calls, it would find **Plan** an easier task. However, we find that a more explicit plan doesn't consistently boost performance. Although **Relative Error | Strict** is the principal metric one would ideally want to optimize for, we do not report means over that metric here since the LLM was only able to compute a Valid Target that was also within 10% of the

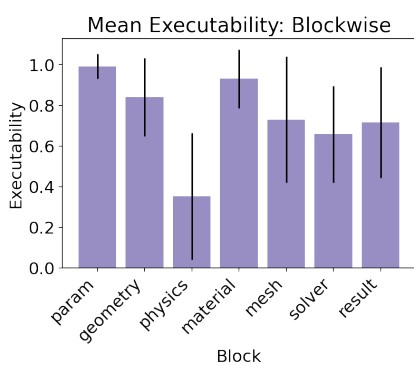

Figure 2: Block-wise executability on 300 samples of code with PhyDoc In-Context and Gemini-1.5-Pro. The physics block has the lowest executability. Error bars are standard deviations.

correct answer for a single problem in the Multi-Turn Agent and ModelSpecs + PhyDoc experiments. For this problem, the correct target value is 18.3° Celsius, and the value exported by the LLM is 20° Celsius, which is a default temperature in COMSOL Multiphysics®: this is an indicator of the solution not being solved correctly. While a stricter relative error threshold would filter out such serendipitous matches, this risks filtering out problems in which a solution might be conceptually correct but differs from the target because of say, differences in solver and mesh sizes. In Figure 2, we used the initial population of samples generated by the LLMs to analyze executability within different blocks of code. The physics block has the lowest executability, thus motivating our focus on metrics that seek to measure performance on the physics decisions (API calls) made by the LLM.

## 5    Discussion

Our benchmark seeks to inspire rigorous evaluations of the capabilities of LLMs on solving problems that require simulating physical phenomena in the real world and performing numerical analysis. Such problems are ubiquitous in science and engineering, and solving them requires synthesizing reasoning over the physics domain with the ability to leverage numerical analysis software such as FEA. Our dataset serves as a novel testbed to evaluate the ability of LLMs to interact with feedback from an execution environment, correctly analyze and attribute errors and correct them. Future directions include devising agent strategies that boost performance, adding more benchmark problems, increasing problem complexity such as working with more complex geometries, and examining whether performance improves by leveraging GUI visual information. An LLM system that can successfully master FEA software would unlock the ability to solve a range of real-world engineering and physics problems.

## 6    Acknowledgements

We are grateful to Eser Aygün for valuable suggestions on agent design. We thank Stephan Hoyer and Marc Coram for useful comments on the draft, and are grateful to Lizzie Dorfman and Rachel Stigler for their guidance. At Harvard, NM is partially supported by the National Science Foundation under Cooperative Agreement PHY2019786 (The NSF AI Institute for Artificial Intelligence and Fundamental Interactions).

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

# Appendix

## A  Dataset Curation

### A.1  Selection Criteria:

We chose tutorials that satisfied the following considerations:

1. *Simpler Geometry:* COMSOL Multiphysics®can be used to analyze the physics of systems involving intricate geometries such as microwaves or transformers. In these cases, in practice, most problems involve importing a pre-built geometry object that might have been built externally using Computer-Aided Design (CAD) software and to then perform the remaining analysis. Since we wanted to explore the ability to solve the problem end-to-end and without requiring imports of derived objects, we restrict ourselves to problems that did not require imports of geometry, or any other files.

2. *Tutorial / Code Simplicity:* We additionally chose problems that did not involve multiple 'Model' JAVA classes and restricted ourselves to tutorial documents with fewer than 20 pages. The first requirement is a consequence of how our connection to the COMSOL Multiphysics®sandbox is set up, and to make the problem easier for the models to attempt to solve. We additionally ensured that the problems were amenable to computing a numerical artifact.

3. *Solving Speed:* We also excluded any problems whose ground truth code took over a minute to solve.

### A.2  Generation Procedure:

Without any modification, the tutorials might export a single value, a table, or not export any target quantity at all, with the final output being qualitative in nature, such as in the form of plots or figures. For our benchmark, however, we specifically wanted every problem to have a numerically verifiable target value, in order for there to be an absolute notion of correctness (i.e. if the code was fully correct, and aligned with the intent of the problem, it should be able to export this value). This also enables easier evaluation of the problems. The following procedure and guidelines were adopted to curate the benchmark:

- For an initial set of 2-3 problems, model specifications and plans were annotated by hand, by an expert user of COMSOL Multiphysics®.

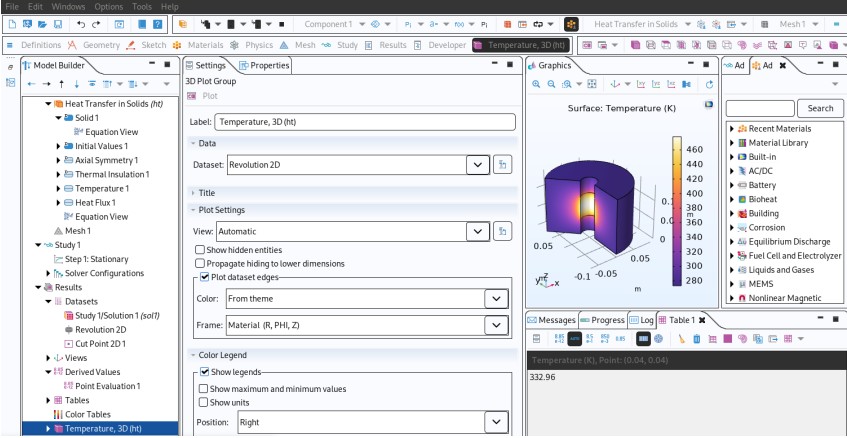

Figure 3: Screenshot of the graphical user interface for the correctly solved problem in Figure 1.

- For subsequent problems, we speed up the benchmark generation procedure by following an initial LLM-assisted data generation process, with the final verification steps involving humans. An LLM is provided with a tutorial, as well as a two-shot prompt with the expert annotated model specifications.

- The LLM is entasked with returning a model specification for the tutorial that has the same format. This requires the LLM to identify an appropriate target value from the tutorial which it does from either the text or the figures, and returning a model specification for computing this target value.

- The LLM is then asked to create a plan corresponding to the model specifications, using a two-shot prompt with two plans. The utility of the tutorials are that the plan is closest to the GUI instructions listed in the tutorial, while model specifications is more concise.

- A ground truth code that can compute the correct value is then generated for the problem. We manually verify that the ground code when run, exports the desired target value. This step also involves simultaneously ensuring that all information required to build the model is contained in the plan, and in the model specifications by editing the LLM-generated drafts and ensuring that no Translation Errors are encountered when parsing and executing the ground truth code in COMSOL Multiphysics®using the bridge described in Appendix C.1 or that any errors if encountered are in non-crucial lines and do not prevent the solution from being computed. Any missing or incorrect information is fixed, and the selection_information field, that contains numerical identities of boundaries and points is also created.

- We add an instruction to export the output to `OUTPUT_PATH/output.txt` in the model specifications and plan.

### A.3   Fields from an example entry:

Here is an example of the information saved for one of the problems, `comsol_453` based on Steady-State 2D Axisymmetric Heat Transfer with Conduction, Heat Transfer Module Application Gallery, COMSOL Multiphysics®v.6.1. COMSOL AB, Stockholm, Sweden, 2023 [21]:
**Model Specifications:**

> ## Finite Element Analysis Description: 2D Axisymmetric Steady-State Heat Conduction in a Cylinder
> **ANALYSIS TYPE:** Steady-state heat conduction with axisymmetric geometry.
> **GEOMETRY:** * The domain is a cylindrical section defined by:
>    - Inner radius: 0.02 m
>    - Outer radius: 0.1 m
>    - Height: 0.14 m

* The geometry represents a 2D cross-section of this cylinder, with the width corresponding to the difference between the inner and outer cylindrical surfaces.

**LOADING:** * A constant heat flux of 5e5 W/m$^2$ is applied to the inner cylindrical surface, between z = 0.04 m and z= 1 m. The remaining portion of the inner cylindrical surface is insulated.

**BOUNDARY CONDITIONS:** * The outer cylindrical surface, top surface, and bottom surface have a uniform temperature of 273.15 [K].

**MATERIAL PROPERTIES:** * Thermal conductivity (k): 52 W/(m·K)

**ELEMENT TYPES:** The analysis can utilize 2D axisymmetric heat transfer elements.

**MESHES:** The default mesh can be used.

**OUTPUT:** The analysis should determine the temperature in Kelvins [K] (Kelvins are the default units) at a specific point on the inner cylindrical surface:

* Radial Coordinate (r): 0.04 m * Axial Coordinate (z): 0.04 m

Export the table with the value to OUTPUT_PATH/output.txt

**Plan:**

## Implementing the 2D Axisymmetric Steady-State Heat Conduction in a Cylinder in COMSOL Multiphysics:

**1. Model Setup:**
* **New Model:** Start COMSOL Multiphysics and create a new model.
* **Space Dimension:** Select "2D Axisymmetric".
* **Physics Interface:** Select "Heat Transfer > Heat Transfer in Solids (ht)".
* **Study Type:** Choose "General Studies > Stationary".

**2. Geometry Definition:**
* **Rectangle:** Create a rectangle representing the cross-section of the cylinder:
* Width: 0.08 m
* Height: 0.14 m
* Corner Position: (r, z) = (0.02, 0) m
* **Point:**
* In the r field, type 0.02 0.02
* In the z field, type 0.04 0.1

**3. Definitions:** * **Boundaries:** Define selections for the following boundaries:
* **Inner Cylinder Surface:** Left edge of the rectangle
* **Outer Cylinder Surface:** Right edge of the rectangle
* **Top Surface:** Top edge of the rectangle
* **Bottom Surface:** Bottom edge of the rectangle

**4. Physics Settings:** * **Heat Conduction, Solid:**
* For the Thermal Conductivity (k), choose User defined, and type 52 W/(m.K).
* Under Thermodynamics Solid, choose User defined for Cp and rho.
* Domain Selection:** Select all domains or Domain 1.
* **Boundary Conditions:**
* **Temperature:**
* In the Temperature section, type 273.15 [K] for T$_0$.
* Select Boundaries 2, 5 and 6.
* **Heat Flux:**
* Apply a "Heat Flux" boundary condition with a constant value q0 of 5e5 W/m$^2$.
* Select Boundary 3.

**5. Meshing:**
* **Mesh Creation:** Use the default mesh.

**7. Study Settings:**
* **Solver Configuration:** Use default solver settings for the "Stationary" study.

**8. Analyzing Results:**

* **Temperature at Target Point:**
* Create a "Cut Point 2D" dataset at this location first and then use that dataset in the point evaluation:
* Locate the Point Data section under Cut Point 2D and type R = 0.04 m, Z = 0.04 m
* Use a "Point Evaluation" feature to evaluate the temperature (in K) at the target point.
* Export the table containing this value to OUTPUT_PATH/output.txt.

**Selection Information:**

DOMAINS: Thermal Conductivity applies to the entire geometry, all domains, or Domain 1.
BOUNDARIES: * The temperature setting T_0 = 273.15 [K] applies to Boundaries 2, 5 and 6.
* The constant heat flux applies to Boundary 3.

**Target Description:** Temperature at the location R = 0.04 m, Z = 0.04 m in K.

**Target Value:** 333

**Target Units:** K

**Ground Truth Code:**

```
model.component().create("comp1", true);

model.component("comp1").geom().create("geom1", 2);
model.component("comp1").geom("geom1").axisymmetric(true);

...
model.component("comp1").physics().create("ht", "HeatTransfer",
    "geom1");
...
model.component("comp1").physics("ht").create("temp1", "
    TemperatureBoundary", 1);
model.component("comp1").physics("ht").feature("temp1").set("T0
    ", "273.15[K]");
...
model.result().table("tbl1").comments("Point Evaluation 1");
model.result().numerical("pev1").set("table", "tbl1");
model.result().numerical("pev1").setResult();
model.result().table("tbl1").save("OUTPUT_PATH/output.txt");
```

**Model Tree:**

```
model
├─ parameters
│   └─ Parameters 1
├─ functions
│   ├─ Analytic
│   ├─ Analytic
│   └─ Blackbody Radiation Intensity
├─ components
│   └─ Component 1
├─ geometries
│   └─ Geometry 1
│       ├─ Rectangle 1
│       ├─ Point 1
│       └─ Form Union
```

```
...
├── physics
│   └── Heat Transfer in Solids
│        ├── Solid 1
│        │   └── Opacity 1
│        ├── Initial Values 1
│        ├── Axial Symmetry 1
│        ├── Thermal Insulation 1
│        ├── Isothermal Domain Interface 1
│        │   └── Layer Opacity 1
│        ├── Local Thermal Nonequilibrium Boundary 1
│        ├── Opaque Surface 1
│        ├── Continuity 1
│        ├── Temperature 1
│        └── Heat Flux 1
...
├── studies
│   └── Study 1
│        └── Stationary
├── solutions
│   └── Solution 1
│        ├── Compile Equations: Stationary
│        ├── Dependent Variables 1
│        │   └── Temperature (comp1.T)
│        └── Stationary Solver 1
│             ├── Direct
│             ├── Advanced
│             ├── Fully Coupled 1
│             ├── Direct, heat transfer variables (ht)
│             └── AMG, heat transfer variables (ht)
│                  └── Incomplete LU
├── batches
├── datasets
│   ├── Study 1//Solution 1
│   └── Cut Point 2D 1
├── evaluations
│   └── Point Evaluation 1
├── tables
│   └── Table 1
├── plots
└── exports
```

# B   Evaluation Details

## B.1   Executability

The LM output is first parsed to identify the block with Java API calls, and further parsed to pythonize the lines (Appendix C.1). This filters out lines that are not code or cannot be pythonized and results in a sequence of COMSOL Multiphysics®API calls and their 'pythonized' counterparts, all of which start with `model.` and end with ';'.

The pythonized lines are then passed to the MPh client, and replies for each line are received. We parse API replies using the following patterns. A reply containing any of the following ['Messages', 'has no attribute', 'No matching overloads', 'invalid syntax', 'Exception', 'is not defined'] are considered Syntax Errors. Replies with ['Ambiguous', 'comma', 'No Model set'] are Translation errors. The last category category is rare in our experiments and are occasionally encountered when we tested adding new problems to the benchmark that contained lines that weren't translated correctly in the query: the first two flag errors in the query to COMSOL Multiphysics®via Mph, while the last indicates that

an action is being done on a non-existent model, which is inconsistent with the setup of the code. All other replies are designated Correct.

$$Executability = \frac{\text{CorrectLines}}{\text{TotalParsedLines}} \tag{1}$$

## B.2 Code Similarity Score

We use the difflib [22] package, that computes a score between 0 and 1 as a measure of string similarity, using the ratio of the lengths of the longest matched subsequences to the ratio of the lengths of strings being compared. Code Similarity reflects this score between the generated code and the ground truth code. It is not surprising that this metric has the least change since significantly different blocks of code might yield the same answer. As a specific example, a `model.study("std1").run();` will leverage COMSOL Multiphysics®'s default numerical solver for the problem. However, this could also be represented explicitly using large blocks of `model.sol("sol1")...` lines in the Ground Truth Code field.

## B.3 Model Tree Score

The model tree representation of the model built by the language model can be extracted, and one can use the same similarity score as above to compute a similarity score relative to the target tree. We expect this to be a more reliable measure of alignment since different blocks of code that build the same model will have the same model tree (addressing the case described in Code Similarity). Using the formula below, the score will be 1.0 if the trees are identical, and 0.0 if the trees are equivalent to a tree before any code is run.

$$ModelTreeScore = \frac{\text{Score(LM, GT)} - \text{Score(Empty, GT)}}{1.0 - \text{Score(Empty, GT)}} \tag{2}$$

The following is an empty tree, corresponding to a model that has only been initialized, before any code is run.

```
model
├─ parameters
│   └─ Parameters 1
├─ functions
├─ components
├─ geometries
├─ views
├─ selections
├─ coordinates
├─ variables
├─ couplings
├─ physics
├─ multiphysics
├─ materials
├─ meshes
├─ studies
├─ solutions
├─ batches
├─ datasets
├─ evaluations
├─ tables
├─ plots
└─ exports
```

## B.4 Valid Target

There are various ways in which computing the correct value and exporting it to a table may fail: a) the LLM's code forgets the export command to the API and no table is exported b) an empty table is exported or, c) a table containing an incorrect value is exported, such as a default value or the wrong quantity (eg: time instead of temperature). Failure modes b) and c) are far more common than a) and occur when the code is not fully correct and the partially constructed COMSOL Multiphysics®model exports nothing or an incorrect value. For instance, a partially solved model that was asked to compute the temperature at time=190s might export a table where the last value was 190 but because of errors in model construction, no temperature was exported. In such a case if the ground truth answer is say, 185°C, without verifying the physical quantity, one would mistakenly evaluate the algorithmically parsed figure 190 to be quite close to the target. In other cases, the software might export a default such as 293.15 K if the solver did not solve correctly.

If a table containing the target quantity is exported, it is first read and parsed. The last value in the table is algorithmically extracted. To address this problem, we ask an LLM (Gemini-1.5-Pro), to extract the exported value and units from the table, if it is a match for the target description, and minimize the chances of incorrectly evaluating these failure modes as valid solutions.

---

**Evaluate Prompt**

You are provided with a table that was exported by a model built in COMSOL. The table * should * contain the EXPECTED TARGET QUANTITY. The following failure modes may occur when the model is not built correctly:
1. The table might be empty or might export a physical quantity that is different from the expected target quantity.
2. The table might export the same physical quantity, but the quantity is just an initial or boundary condition, or a default value that was exported, instead of the result of genuinely numerically solving the problem. You can find numbers already in the problem description in 'PROBLEM'. Default values include 20degreesCelsius, 293.15 K, 0 etc.
Carefully examine the 'TABLE' and compare it with the units and description of the expected target quantity and the numbers in 'PROBLEM' to assess whether the table exported a value that was the result of genuinely numerically solving the problem. You must return TARGET VALUE and TARGET UNITS in json format if the table was the result of genuinely solving the model, computing a solution and exporting it. Return 'N/A' for both fields if the table suffers from either of the failure modes described above.

———

**PROBLEM: {{ problem_description }}**

———

**EXPECTED TARGET QUANTITY: {{ target_description }}**

**TABLE: {{ table }}**

**REPLY:**

---

We then compute the number of problems for which the LM was able to parse the reply and convert it to a JSON. This fraction is the number we report as Valid Target.

## B.5 Relative Error | Strict

Our strict filter for whether a model has truly solved the problem is to take the subset of problems for which the problem was judged to be a valid export by the LLM, and to consider the algorithmically parsed last value. We then compute the relative error of this value against the ground truth target value. If this value is less than 10%, we consider it valid.

### B.6 Physics Metrics

The interface lines are parsed from the ground truth code by finding lines that fit the regex pattern for interface creation. Likewise for the feature creation and feature property modification lines. Each of these lines of codes can be considered as an "Action" consisting of an Action Type (eg: Create Interface) with corresponding Arguments (eg: Interface tag, Name of the Interface, Geometry).

**Create Interface:** `model.component("comp1").physics().create("Interface_tag", "InterfaceName", "Geometry_tag");`

Eg: `model.component("comp1").physics().create("ht", "HeatTransfer", "geom1");`

**Create Feature:** `model.component("comp1").physics("Interface_tag").create("Feature_tag", "FeatureName", Dimension);`

Eg: `model.component("comp1").physics("ht").create("temp1", "TemperatureBoundary", 1);`

**Modify Feature Property:** `model.component("comp1").physics("Interface_tag").feature("Feature_tag").set("Param", "Value");`

Eg: `model.component("comp1").physics("ht").feature("temp1").set("T0", "1000[degC]");`

#### B.6.1 Interface Factuality

We check whether the Interface name exists in a list of known COMSOL Multiphysics®interfaces. If it exists in this list, we assign it a factuality of 1, else 0.

#### B.6.2 Interface Recall

How many GT interface creation actions (ignoring Interface_tag) were also in the LM code? This checks whether the same interface was defined on the same geometry. 'nan' if there are no interfaces in the GT (not encountered in our dataset).

#### B.6.3 Feature Recall

Since multiple features may be created under the same interface (eg: 2 Boundary Conditions with different temperatures), we compute the occurrences of *each* GT feature name in the GT code and in the LM code, and a recall for each GT feature name, and then average over all GT features. In our implementation, if no GT features are defined, a) AND no LM features are defined the recall is 1, b) but LM features are defined, the recall is 0.

#### B.6.4 Feature Dimension

Let $F_c$ be all the GT features that are also created by the LM solution. Let $Dim_c$ be the set of $F_c$ such that the LM feature has the same dimension as the GT feature. Feature Dimension = $\frac{|Dim_c|}{|F_c|}$

This is a correctness and physics reasoning metric as opposed to an alignment-focused metric since creating a TemperatureBoundary with dimension 2 attempts to create a 2D temperature boundary condition. Creating a TemperatureBoundary with dimension 1 attempts to create a temperature on an edge. Thus this measures the LM's ability to correct deduce the spatial dimension of boundary conditions or other features from the context of the problem.

#### B.6.5 Feature Property Recall

This compares the modify feature property actions. It computes how many GT modify feature property actions were also in the ground truth, *ignoring* differences in Interface_tag and Feature_tag. If no GT properties are modified, a) AND no LM features are modified the recall is 1, b) but LM features are modified, the recall is 0.

# C    Querying the COMSOL Multiphysics®API from Python

## C.1    The Python-COMSOL Multiphysics®Bridge

The raw output of the LLM is a string containing COMSOL Multiphysics®API commands in Java. An interface between Python and COMSOL Multiphysics®is needed to execute this code and interact in other ways with the API. We use the Python package MPh [23] and Rpyc for this. MPh is a scripting interface built on JPype [24] that enables a Python program to communicate with and build a model in COMSOL Multiphysics®. Each Java API command in the LM's output can be 'pythonized' algorithmically. In most cases, the pythonized line is near identical to the Java line. However, due to differences in Java and Python syntax there exist some corner cases that need to be handled separately. Eg: 'new String[]' is exclusively a Java construction, while the notation for booleans in Python is True / False as opposed to true / false in Java. Thus a 'pythonizer' is constructed that parses and translates Java API calls to their Python counterparts.

The setup involves the following assumptions: an MPh client object is created. This behaves like a stateful 'sandbox', where models can be built by LLMs, code can be evaluated, or information such as the current state of the model tree, properties under a node and the exported table can be queried and retrieved. Although multiple models can be created and set under the client, for simplicity we work with settings that involve a single model. Before running a new solution, the existing model is deleted and a new blank model is created. The LLM actions will modify this blank model. Thus, by design, all lines of code the LLM outputs, should start with 'model.' and end with ';'.

## C.2    COMSOL Multiphysics®Code Structure

1. *Geometry*, if any: This involves identifying the dimensionality of the problem, and constructing a representation of the object being modelled, say a cup, by creating and composing primitive shapes such as ellipses or rectangles to build the object. While already constructed geometries can also be imported from other software such as CAD, in our benchmark, we currently restrict ourselves to models for which we construct the geometry from scratch in COMSOL. This typically starts with a 'model.component("comp1").geom' pattern.

2. *Physics*:  This will include specifying all the physical conditions for the problem, including initial or boundary conditions, forces, properties or in the case of mathematics problems, the differential equation.  This typically starts with a 'model.component("comp1").physics' pattern. Some problems may additionally have lines that begin with 'model.component("comp").multiphysics', and set up the coupling between different kinds of physical phenomena. We categorize these lines, if any as 'physics' in Figure 2.

3. *Material*: Creating materials and assigning them to domains. One can either assign known materials such as 'Copper' and the object will inherit the default properties of that material, or define a blank material and its properties such as conductivity from scratch. This typically starts with a 'model.component("comp1").material' pattern.

4. *Mesh*: Usually a shorter step that involves meshing the surfaces of the geometry to set up elements. This typically starts with a 'model.component("comp1").mesh' pattern.

5. *Study / Solver:* This involves specifying the conditions of the analysis and solver, such as the number of timesteps. While the solver code can be modified to override defaults, COMSOL also has the ability to automatically populate the model with the default solvers most apt for a given problem. This typically starts with a 'model.study' or 'model.sol' pattern respectively. In Figure 2, we categorize both patterns as 'solver'.

6. *Results:* Once the numerical solver has completed the analysis, one will likely postprocess the problem, in order to generate desired plots or tables.  This typically starts with a 'model.result' pattern.

# D    Agent Design

Our single-turn results, particularly on executability and hallucinated interfaces, highlight the need to ground the LLM's responses with feedback from and knowledge about how to use the API. Given

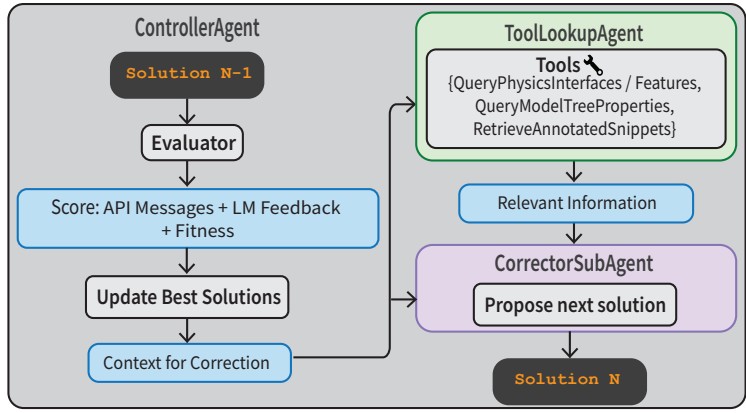

Figure 4: An overview of the agent and environment design, and the steps involved returning the next solution.

a single try, the LLM is likely to make some errors, and we hypothesize that receiving feedback informative of the nature and location of the errors will allow it to correct those errors in a directed fashion. We thus build an environment in which the LLM can interact with the API, receive feedback and attempt to correct its solution. We design a nested multi-agent system that interacts with the COMSOL Multiphysics®API, as well as tools (or specialized functions) that contain information relevant to correcting a problem. Each agent has a specific role and input context.

**Design Principles** (1) To minimize failures or longer-than-desired chains of calls, we adopt an algorithmic sequence of agent calls *except* when we call tools. (2) High executability does not guarantee alignment or correctness, since API messages alone are not informative about inconsistencies with the problem description, such as incorrect units. We thus combine an LLM's feedback with the API messages to flag such inconsistencies. (3) Calling tools may fail in cases where the argument to the tool is invalid or the call sequence could not be parsed. In such cases we ensure the agent system can still attempt to correct the code, without the relevant information from tool calls.

CONTROLLERAGENT: The main agent that tries to solve the problem description by generating solutions, interacting with the API and calling subagents.
**Input Context**: Problem description.
**Components**: Evaluator, ControllerSubAgent
**Working**: This samples an initial population of N(=20) solutions using PhyDoc In-Context. Each solution is evaluated by the Evaluator. A fitness score, between 0 and 2, is computed for each solution, using the following formula: Executability + ExportSuccessful where ExportSuccessful is 1 if (the solution computed a value AND had executability above 90%) and 0 if not. The controller agent tracks a set of best replies using their fitness. The set of best replies stores at least B(=1) solution, as well as all solutions that successfully computed a value. This agent also determines the context to be sent to the CorrectorSubAgent, using the following algorithm:

- Solution to iterate on: We use an iteration criterion inspired by the Markov Chain Monte Carlo (MCMC) acceptance criterion. The solution to iterate on (rendered in the prompt to the CorrectorSubAgent as "CURRENT CODE") is (a) the last solution if the last solution has equal fitness as the best solution, and (b) the last solution if a random float between [0, 1] is less than $\alpha = \frac{Last\_Fitness}{Best\_Fitness}$, else the best solution.
- ExecutionHistory: The best solutions, if not already used in context upto a maximum of 3 best solutions, in addition to the last N_bad(=1) replies, if not already in context.

EVALUATOR: This returns the feedback for a solution in a 'score' dictionary (Left panel, Figure 5)
**Input Context**: A solution.
**Working**: The evaluator combines execution feedback with subjective feedback from an LLM. The evaluator executes the parsed solution in the API, and stores the API messages as well as the Executability. If Executability exceeds 90%, the VerifierLLM examines the code as well as the

computed value in 'output.txt' and returns subjective feedback on changes that need to be made. To facilitate this, the VerifierLLM sets an analytical guess for what the solution should be at the onset of the problem, and compares the computed value with this guess when it evaluates a solution. The API feedback provides a signal on *syntactical correctness* and the VerifierLLM provides a signal on *completeness*. Note, this evaluator is *not* aware of the GT target value.

CORRECTORSUBAGENT: This returns an updated solution.
**Input Context**: Problem description, Current Code and Feedback, Execution History
**Components**: ToolLookupAgent
**Working**: This calls the ToolLookupAgent and retrieves its reply. It then includes this reply to the rest of the context received from the ControllerAgent to propose the next solution.

TOOLLOOKUPAGENT: This calls tools and returns the information retrieved from them.
**Input Context**: Feedback
**Components**: ToolRegistry
**Working**: The LLM is shown tool descriptions and the input context and must return a list of tool calls, as structured classes using the Langfun [25] package consisting of the tool name and its arguments. If successfully parsed, each tool is called with its arguments and the replies are concatenated (see Figure 5 for the feedback and reply for a single step). The tools in the registry are:

1. QUERYPHYSICSINTERFACES: This returns a list of valid physics interfaces.
2. QUERYPHYSICSFEATURES: This returns the features under an argument *interface* or a list of known features under interfaces.
3. QUERYMODELTREEPROPERTIES: The LLM must call this tool with a *path* argument ('/physics/Heat Transfer in Solids/Solid 1' in Figure 5) to receive the properties under the node corresponding to path.
4. RETRIEVEANNOTATEDSNIPPETS: To call this tool, the LLM must specify a *branch* – one of the conceptual blocks such as physics or geometry – and a *query* – a brief natural language description of a specific step. In Figure 5, the LLM first called this tool with the branch 'geometry' and the query 'Create a 2D axisymmetric geometry in...'. A retriever then looks up the annotated library and retrieves 3 annotations along with their code snippets, most similar to the query made. Thus, this allows the LLM to search a library of code snippets to find the correct ways to express certain steps in code, simulating how a human unfamiliar with a coding language would look up similar examples of code.

At the end of this experiment, the CONTROLLERAGENT saves its best solutions as well as other intermediate states. During evaluation, the best solutions are read in and evaluated. If there are multiple best solutions (in cases where multiple solutions were able to compute a target value), the best solution is the one that maximizes the following formula: Executability + bool(Computed Value) + [(1.0 - Target Relative Error) if (Target Relative Error<1) AND (Valid Target) else 0]. The three conditions together prioritize solutions that (1) had high executability, (2) were complete enough to export any value, albeit incorrect or the wrong quantity and, (3) exported a Valid Target within 10% of the desired value.

# E   Agent Details

## E.1   Tools

### E.1.1   QueryModelTreeProperties

In order to help the LLM learn how to appropriately format a valid path, say to the 'Solid' feature, the current state of the model tree is shown to the ToolLookupAgent LLM. It also has a history of unsuccessful (incorrectly formatted) paths in previous queries to this tool, in order to minimize the chances of incorrectly calling this tool with an invalid path.

### E.1.2   RetrieveAnnotatedSnippets

We use the Discovery Engine API [26] with the model name 'semantic-ranker-512-003' to rank and retrieve the top 3 annotations most similar to the query snippet. The annotation library was generated by taking tutorials and splitting them into code blocks using the patterns described in C.2. There are

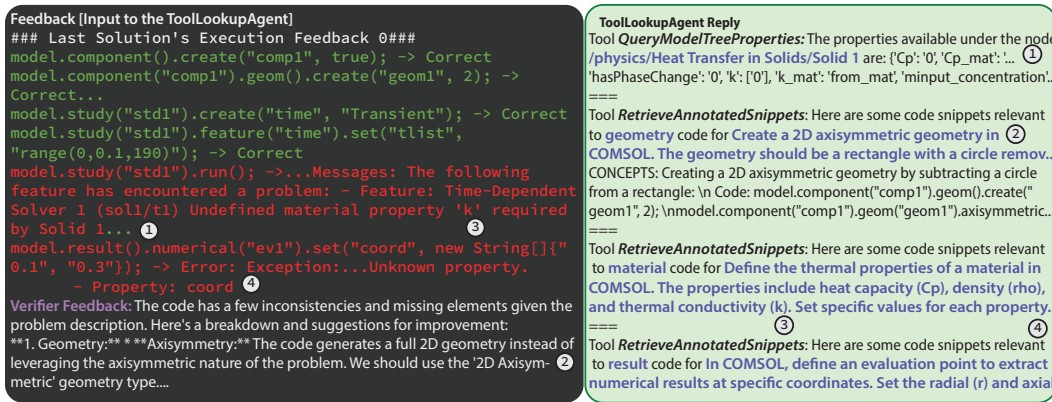

Figure 5: The Evaluator's feedback (*left*) is passed to the ToolLookupAgent, that calls **tools** and returns their concatenated output. Violet, on the *left* indicates that the Verifier Feedback is returned by an LLM, and on the *right*, denotes the arguments chosen by the ToolLookupAgent to call the tools with. The numerical annotations highlight the correspondence between the errors and arguments.

768 pairs of annotations and snippets across all branches of code. Here is an example of an annotation 'summary' and its snippet:

**Summary:** Defining a transient study with a time range from 0 to 0.025 seconds with a step of 1 second. The study will solve for the "spf" physics interface, and a relative tolerance of 0.001 will be used. The number of solver iterations will be automatically determined based on the time step.

**Code:**

```
model.study().create("std1");
model.study("std1").create("time", "Transient");
model.study("std1").feature("time").setSolveFor("/physics/spf", true);
model.study("std1").feature("time").set("tlist", "range(0,0.025,1)");
...
model.study("std1").feature("time").set("solnum", "auto");
```

## E.2 Analysis

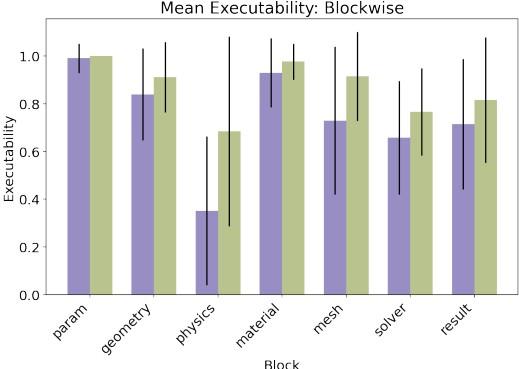

Figure 6: Block-wise executability across the 300 initial samples of code (purple) with PhyDoc In-Context and in the best solution (green) across all problems. Error bars denote standard deviations.

Figure 6 depicts the blockwise executability in the initial sample relative to the best solution across problems. The standard deviations in the best case are higher since we have 1 best solution for each problem, and 20 samples per problem in the initial population. Figure 7 plots the Executability as well as the number of errors over solution iteration. The evolution of the metrics isn't monotonic and in some cases the agent gets stuck on the same solution for some iterations, or takes an incorrect

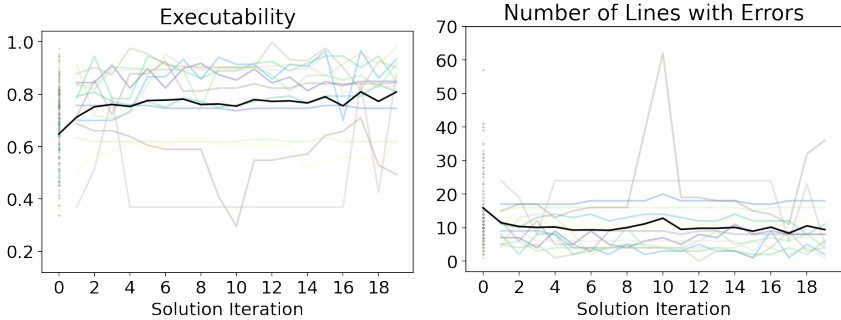

Figure 7: Executability and number of errors over solutions returned by the ControllerAgent. The scatter at 0 denotes the spread from the samples in the initial population and the black line denotes the mean value for the metric at that state across all problems. Each colored line demarcates a different problem.

turn. We added the acceptance criterion to minimize the number of iterations required to "escape" an incorrect turn.

# F   Qualitative Analysis

In Figure 8, we delve into the differences between the LLM-generated code for the **ModelSpecs** task in the baseline (one-shot) setting with Gemini-1.5-Pro, relative to the ground truth code, for the problem in Appendix A.3. At a high-level, the LLM's solution consists of API calls that qualitatively posess the same structure and grammar as in the GT code. This is a problem that requires the LLM to represent the cylindrical cross-section of a cylinder as a rectangle in 2D, with the axisymmetric condition applied for rotational symmetry about the cylinder's axis. The LLM instead creates a 3D geometry and attempts to create a rectangle. This doesn't work as is indicated by the error message, since the rectangle is a 2D construct and cannot be directly created in 3D. Since the rectangle creation action fails, no 'r1' node is created, and any subsequent actions that act on the 'r1' node cannot be executed. This pattern of non-executability is also observed downstream, where all actions on the 'ht' node are rendered invalid because the 'ht' node could not be created in the first place. Note, if the LLM had chosen a 2D geometry, or a 2D axisymmetric geometry, the geometry lines of code would have been correct. The reason they fail is because of an incorrect (3D geometry) decision taken first.

Next, the LLM chooses to set the thermal conductivity under the materials node. These lines of code are executable and this may be a valid choice, if the physics node is properly able to query properties redefined under the materials node.

The LLM tries to create a 'HeatTransferinSolids' interface. This is a subtle error. Heat Transfer in Solids is indeed the correct *natural language* name for this interface under COMSOL Multiphysics®and is often referred to as such in documentation on the internet. However, this is *not* the correct *syntactical* name for the interface, which, as can be seen in the GT code, is 'HeatTransfer'. Errors like these are likely why the adding the list of physics interfaces and features to the prompt (*PhyDoc In-Context*) improve performance on both tasks. Since the LLM's chosen interface and features differ from the ground truth in this example, the Interface Recall and Feature Recall metrics are both 0, as is the Interface Factuality metric (since 'HeatTransferinSolids' does not exist). The GT code modifies 5 features, of which the LLM only modifies 1 (setting $T_0$ to 273.15 K). Thus the Modify Feature Property score is 0.2.

In the results section, the model incorrectly attempts to set the properties 'geom', 'x' and 'y; under the point evaluation node. All three lines trigger 'Unknown property' exceptions.

```
Ground Truth Code
model.component().create("comp1", true);
model.component("comp1").geom().create("geom1", 2);
model.result().table().create("tbl1", "Table");
model.component("comp1").geom("geom1").axisymmetric(true);
model.component("comp1").mesh().create("mesh1");
model.component("comp1").geom("geom1").create("r1", "Rectangle");
model.component("comp1").geom("geom1").feature("r1").set("pos",
new double[]{0.02, 0});
model.component("comp1").geom("geom1").feature("r1").set("size",
new String[]{"0.08 [m]", "0.14"});
model.component("comp1").geom("geom1").create("pt1", "Point");
model.component("comp1").geom("geom1").feature("pt1").set("p", new
String[]{"0.02 0.02", "0.04 0.1"});
model.component("comp1").geom("geom1").run();
model.component("comp1").physics().create("ht", "HeatTransfer",
"geom1");
model.component("comp1").physics("ht").create("temp1",
"TemperatureBoundary", 1);
model.component("comp1").physics("ht").feature("temp1").selection(
).set(2, 5, 6);
model.component("comp1").physics("ht").create("hf1",
"HeatFluxBoundary", 1);
model.component("comp1").physics("ht").feature("hf1").selection().
set(3);
model.result().table("tbl1").comments("Point Evaluation 1");
model.component("comp1").physics("ht").feature("solid1").set("k_ma
t", "userdef");
model.component("comp1").physics("ht").feature("solid1").set("k",
new int[][]{{52}, {0}, {0}, {0}, {52}, {0}, {0}, {0}, {52}});
model.component("comp1").physics("ht").feature("solid1").set("rho_
mat", "userdef");
model.component("comp1").physics("ht").feature("solid1").set("Cp_m
at", "userdef");
model.component("comp1").physics("ht").feature("temp1").set("T0",
"273.15[K]");
model.component("comp1").physics("ht").feature("hf1").set("q0_inpu
t", "5e5");
model.study().create("std1");
model.study("std1").create("stat", "Stationary");
<model.sol block>
model.result().dataset().create("rev1", "Revolve2D");
model.result().dataset().create("cpt1", "CutPoint2D");
model.result().numerical().create("pev1", "EvalPoint");
model.result().numerical("pev1").set("probetag", "none");
<model.sol block>
model.result().dataset("rev1").label("Revolution 2D");
model.result().dataset("rev1").set("startangle", -90);
model.result().dataset("rev1").set("revangle", 225);
model.result().dataset("cpt1").set("pointx", 0.04);
model.result().dataset("cpt1").set("pointy", 0.04);
model.result().numerical("pev1").set("data", "cpt1");
model.result().numerical("pev1").set("table", "tbl1");
model.result().numerical("pev1").setResult();
model.result().table("tbl1").save("OUTPUT_PATH/output.txt");
```

```
LLM Reply
model.component().create("comp1", true);
model.component("comp1").geom().create("geom1", 3);
model.component("comp1").mesh().create("mesh1");
model.component("comp1").geom("geom1").create("r1", "Rectangle");
-> Messages: The requested geometry operation is unknown or
        cannot be created in this context.
model.component("comp1").geom("geom1").feature("r1").set("pos",
new double[]{0.02, 0}); -> Messages: Unknown feature. Tag: r1
model.component("comp1").geom("geom1").feature("r1").set("size",
new double[]{0.08, 0.14}); -> ""
model.component("comp1").geom("geom1").run();
model.component("comp1").material().create("mat1", "Common");
model.component("comp1").material("mat1").propertyGroup().create(
"def", "Constants");
model.component("comp1").material("mat1").propertyGroup("def").se
t("thermal_conductivity", "52[W/(m*K)]");
model.component("comp1").physics().create("ht",
"HeatTransferInSolids", "geom1"); -> Messages: Unknown physics
        interface.
model.component("comp1").physics("ht").feature("temp1").set("T0",
"273.15[K]"); -> Messages: Unknown feature. Tag: ht
model.component("comp1").physics("ht").create("hf1", "HeatFlux",
2); -> ""
model.component("comp1").physics("ht").feature("hf1").selection()
.set(3); -> ""
model.component("comp1").physics("ht").feature("hf1").set("q0",
"5e5[W/m^2]"); -> ""
model.component("comp1").mesh("mesh1").create("map1", "Map");
model.component("comp1").mesh("mesh1").feature("size").set("hauto
", 4);
model.component("comp1").mesh("mesh1").run();
model.study().create("std1");
model.study("std1").create("stat", "Stationary");
model.study("std1").feature("stat").setSolveFor("/physics/ht",
true); -> Messages: /physics/ht is not a valid path of a model
        entity.
model.study("std1").run();
model.result().numerical().create("pev1", "EvalPoint");
model.result().numerical("pev1").set("expr", new String[]{"T"});
model.result().numerical("pev1").set("unit", new String[]{"K"});
model.result().numerical("pev1").set("descr", new
String[]{"Temperature"});
model.result().numerical("pev1").set("geom", "geom1"); ->
Messages: Unknown property. Property: geom
model.result().numerical("pev1").set("x", 0.04); -> Messages:
        Unknown property. Property: x
model.result().numerical("pev1").set("y", 0.04); -> Messages:
        Unknown property. Property: y
model.result().table().create("tbl1", "Table");
model.result().table("tbl1").comments("Point Evaluation 1");
model.result().numerical("pev1").set("table", "tbl1");
model.result().numerical("pev1").setResult();
model.result().table("tbl1").save("OUTPUT_PATH/output.txt");
```

Figure 8: Comparing a Ground Truth code with the LLM-generated code. The ochre-colored lines or arguments in the GT code demarcate lines or arguments that were modified or absent in the LLM generated code. The green (red) lines in the LLM-generated code denote lines that were (were not) executable. The arrows against the red lines show the abbreviated API message returned for each non-executable line.

# G   Prompts

## G.1   Single Query Prompts

**ModelSpecs** | One-Shot

You are an experienced COMSOL engineer. You must solve the problem to compute the desired TARGET QUANTITY by generating COMSOL JAVA API code. The model creation line "'Model model = ModelUtil.create("Model");"' has already been generated and you should not repeat this line. All lines of code must begin with 'model.'

You must not generate any 'model.sol...' solver code but should ensure that your 'model.study...' block ends with a 'model.study("std1").run();'. This will automatically create and run the default solver for the problem. Use the example provided below to infer how to format your response and generate COMSOL code. ===
EXAMPLE 0:
**PROBLEM DESCRIPTION:** ## Stress Analysis of an Elliptic Membrane
**ANALYSIS TYPE:**
* Linear elastic, Plane Stress.
**GEOMETRY:**
* The domain is a quarter of an elliptical membrane.

* The outer curved edge is defined by the equation: $(x/3.25)^2 + (y/2.75)^2 = 1$
* The inner curved edge is defined by the equation: $(x/2)^2 + y^2 = 1$
* Thickness: 0.1 meters (uniform throughout)
* Labeled points: * Bottom Left Corner, Point O: (x = 2.0, y= 0)
**LOADING:**
* Uniform outward pressure of 10 MPa is applied on the outer curved edge, normal to the boundary. * The inner curved edge is unloaded.
**BOUNDARY CONDITIONS:**
* Left Edge: Symmetry about the y-axis, implying zero displacement in the x-direction. * Bottom Edge: Symmetry about the x-axis, implying zero displacement in the y-direction.
**MATERIAL PROPERTIES:**
* Isotropic: The material properties are the same in all directions. * Young's Modulus (E): 2.1 x $10^1$1 Pa * Poisson's Ratio ($\nu$): 0.3
**ELEMENT TYPES:**
* Plane stress: The analysis assumes the membrane is thin and subjected to in-plane loading.
**MESHES:**
* A mapped quadrilateral mesh over the entire Quarter-Symmetry Domain.
**OUTPUT:**
* The analysis aims to calculate the tangential edge stress ($\sigma_{yy}$) at point O.
Export the table containing this value to OUTPUT_PATH/output.txt.
**SELECTION IDENTITIES:**
DOMAINS: Quarter-symmetry membrane: Domain 1
BOUNDARIES: Left edge, Symmetry Boundary Condition: Boundary 1
Bottom edge, Symmetry Boundary Condition: Boundary 2
Outer Curved Edge, Boundary Load: Boundary 4
POINTS:
O: Point 3
TARGET QUANTITY to be computed: Tangential stress ($\sigma_{yy}$) at point O in Pa. Export the table containing this value to OUTPUT_PATH/output.txt.
**ANSWER:**

```java
model.component().create("comp1", true);
model.component("comp1").geom().create("geom1", 2);
model.component("comp1").mesh().create("mesh1");

model.component("comp1").geom("geom1").create("e1", "Ellipse");
model.component("comp1").geom("geom1").feature("e1").set("
    semiaxes", new double[]{3.25, 2.75});
model.component("comp1").geom("geom1").feature("e1").set("angle
    ", 90);
model.component("comp1").geom("geom1").create("e2", "Ellipse");
model.component("comp1").geom("geom1").feature("e2").set("
    semiaxes", new int[]{2, 1});
model.component("comp1").geom("geom1").feature("e2").set("angle
    ", 90);
model.component("comp1").geom("geom1").create("dif1", "
    Difference");
model.component("comp1").geom("geom1").feature("dif1").selection
    ("input").set("e1");
model.component("comp1").geom("geom1").feature("dif1").selection
    ("input2").set("e2");
model.component("comp1").geom("geom1").run();
model.component("comp1").geom("geom1").run("fin");

model.component("comp1").selection().create("sel1", "Explicit");
model.component("comp1").selection("sel1").geom("geom1", 0);
model.component("comp1").selection("sel1").set(3);
model.component("comp1").selection("sel1").label("O");
```

```
model.component("comp1").material().create("mat1", "Common");
model.component("comp1").material("mat1").propertyGroup().create
    ("Enu", "Young's modulus and Poisson's ratio");
model.component("comp1").material("mat1").propertyGroup("Enu").
    set("E", "2.1e11");
model.component("comp1").material("mat1").propertyGroup("Enu").
    set("nu", "0.3");

model.component("comp1").physics().create("solid", "
    SolidMechanics", "geom1");
model.component("comp1").physics("solid").create("sym1", "
    SymmetrySolid", 1);
model.component("comp1").physics("solid").feature("sym1").
    selection().set(2);
model.component("comp1").physics("solid").create("sym2", "
    SymmetrySolid", 1);
model.component("comp1").physics("solid").feature("sym2").
    selection().set(1);
model.component("comp1").physics("solid").create("bndl1", "
    BoundaryLoad", 1);
model.component("comp1").physics("solid").feature("bndl1").
    selection().set(4);

model.component("comp1").physics("solid").prop("Type2D").set("
    Type2D", "PlaneStress");
model.component("comp1").physics("solid").prop("d").set("d",
    ".1");
model.component("comp1").physics("solid").feature("bndl1").set("
    LoadType", "FollowerPressure");
model.component("comp1").physics("solid").feature("bndl1").set("
    FollowerPressure", "-10 [MPa]");

model.component("comp1").mesh("mesh1").create("map1", "Map");
model.component("comp1").mesh("mesh1").feature("size").set("
    hauto", 4);
model.component("comp1").mesh("mesh1").run();

model.study().create("std1");
model.study("std1").create("stat", "Stationary");
model.study("std1").feature("stat").setSolveFor("/physics/solid
    ", true);
model.study("std1").run();

model.result().numerical().create("pev1", "EvalPoint");
model.result().numerical("pev1").selection().named("sel1");
model.result().numerical("pev1").set("probetag", "none");

model.result().table().create("tbl1", "Table");
model.result().table("tbl1").comments("Point Evaluation 1");
model.result().numerical("pev1").set("table", "tbl1");
model.result().numerical("pev1").set("expr", new String[]{"solid
    .syy"});
model.result().numerical("pev1").set("unit", new String[]{"N/m
    ^2"});
model.result().numerical("pev1").set("descr", new String[]{"
    Stress tensor, yy-component"});
model.result().numerical("pev1")
       .set("const", new String[][]{{"solid.refpntx", "0", "
            Reference point for moment computation, x-coordinate"},
            {"solid.refpnty", "0", "Reference point for moment
            computation, y-coordinate"}, {"solid.refpntz", "0", "
```

```
            Reference point for moment computation, z-coordinate
            "}});*
model.result().numerical("pev1").setResult();
model.result().table("tbl1").save("OUTPUT\_PATH/output.txt");
```

===

Now generate the JAVA API code to compute the target quantity for the problem below.
Export the table containing the target quantity to OUTPUT_PATH/output.txt.

**PROBLEM DESCRIPTION: {{problem_description}}**

**TARGET QUANTITY to be computed: {{target_description}}**
s
**ANSWER:**

---

We used the prompt above for the **ModelSpecs** experiment with Gemini-1.5-Pro. We used an identical prompt for Claude-3.5-Sonnet and GPT-4o with the 3rd last line of code in the One-Shot example (marked by *) deleted, since it prevented us from querying those LLMs. We also used the version of the prompt with this line deleted for the Agent experiment using Gemini-1.5-Pro.

**Plan** | One-Shot

You are an experienced COMSOL engineer. You must generate the COMSOL API code in JAVA to execute the steps described in the plan below to compute the desired TARGET QUANTITY by generating COMSOL JAVA API code. The model creation line "'Model model = ModelUtil.create("Model");'" has already been generated and you should not repeat this line. All lines of code must begin with 'model.' You must not generate any 'model.sol...' solver code but should ensure that your 'model.study...' block ends with a 'model.study("std1").run();'. This will automatically create and run the default solver for the problem.
Use the example provided below to infer how to format your response and generate COMSOL code.
===
EXAMPLE 0:
**PLAN:** ## Implementing the Elliptic Membrane Analysis in COMSOL Multiphysics:
**1. Model Setup:**
* **New Model:** Start COMSOL Multiphysics and create a new model.
* **Space Dimension:** Select 2D for the space dimension.
* **Physics Selection:** Choose the "Structural Mechanics Module" and select "Solid Mechanics" as the physics interface.
* **Study:** Create a new "Stationary" study.
**2. Geometry Creation:**
* **Geometry Primitives:** Use the "Ellipse" tool to create two quarter ellipses representing the outer and inner boundaries. To get a quarter-symmetry geometry, limit the sector angle to 90 degrees.
* Outer Ellipse: Center (0, 0), Semi-axes (3.25, 2.75) meters, sector angle = 90 degrees.
* Inner Ellipse: Center (0, 0), Semi-axes (2, 1) meters, sector angle = 90 degrees.
* **Boolean Operations:** Use the "Difference" operation to subtract the inner ellipse from the outer ellipse, creating the quarter-symmetry membrane geometry.
**3. Definitions:**
* **Points:** Create an explicit selection for Point O (Point 3).
**4. Material Properties:**
* **Material Definition:** In the "Material" node, define a new material with the following properties:
* Young's Modulus (E): 2.1e11 Pa
* Poisson's Ratio ($\nu$): 0.3

**5. Physics:**
* **2D Approximation:** Use the "Plane Stress" physics approximation, with a thickness of 0.1 meters.
**6. Boundary Conditions:**
* **Symmetry:** * Select the bottom edge (Boundary 2) and apply a "Symmetry" boundary condition.
* Repeat the same for the left edge (Boundary 1).
* **Pressure Load:** Pressure load of 10e6 Pa acting outwards. * Select the outer curved edge Boundary 4 and apply a "Boundary Load" boundary condition with a "Pressure load" of magnitude of -10 MPa.
**7. Meshing:** * **Mesh Creation:** Right-click on the "Mesh" node and choose "Mapped". * **Mesh Size:** Adjust the mesh size settings to "Fine".
**8. Study Setup:** * **Study Type:** Choose a "Stationary" study to analyze the static equilibrium state. * **Solver Configuration:** Use the default solver settings.
**9. Solving the Model:** * **Compute:** Click on the "Compute" button to run the finite element analysis.
**10. Post-Processing:** * **Point Evaluation:** * Add a "Point Evaluation" node to extract the tangential stress ($\sigma_{yy}$) at point O. * Select point O. * Evaluate the expression "solid.syy". * Export the table containing this value to OUTPUT_PATH/output.txt.
**TARGET QUANTITY to be computed:** Tangential edge stress $\sigma_{yy}$) at O in Pa.
**ANSWER:**

```java
java```
<<SAME AS CODE IN MODELSPECS ONE-SHOT PROMPT>>
```
```

===
Now generate the JAVA API code to compute the target quantity for the problem below, by following the plan described. Export the table containing the target quantity to OUTPUT_PATH/output.txt.

**PLAN: {{problem_description}}**
**TARGET QUANTITY to be computed: {{target_description}}**
**ANSWER:**

We used the prompt above for the **Plan** experiment on Gemini-1.5-Pro

---

**ModelSpecs +Phy-Doc**

You are an experienced COMSOL engineer. You must solve the problem to compute the desired TARGET QUANTITY by generating COMSOL JAVA API code. The model creation line "`Model model = ModelUtil.create("Model");`" has already been generated and you should not repeat this line. All lines of code must begin with `model.` You must not generate any `model.sol...` solver code but should ensure that your `model.study...` block ends with a `model.study("std1").run();`. This will automatically create and run the default solver for the problem.
You are provided with the list of valid physics interfaces and valid features under interfaces. You must only use the interfaces in the available interfaces list.
===
AVAILABLE COMSOL PHYSICS INTERFACES:

```
['BeamCrossSection', 'PorousMediaFlowRichards', '
    MoistureTransportInBuildingMaterials', 'CreepingFlow', '
    CathodicProtection'... <List of 140 Interface>...'
    LumpedBattery', 'CompressiblePotentialFlow', '
    BatteryBinaryElectrolyte', 'ColdPlasma', 'LaplaceEquation', '
    DilutedSpeciesInPorousCatalysts']
```

AVAILABLE FEATURES UNDER INTERFACES:

```
{'ElectromagneticWavesBeamEnvelopes': {'features': ['
    MatchedBoundaryCondition', 'SymmetryPlane', 'Scattering', '
    TransitionBoundaryCondition', 'Impedance', 'Port', '
    FieldContinuity'], 'physics_tags': ['ewbe']}, '
    TransientPressureAcoustics': {'features': ['InteriorSoundHard
    ', 'InteriorLumpedSpeakerBoundary', '
    TransientMonopoleLineSource', 'CylindricalWaveRadiation', '
    Impedance', 'NonlinearAcousticsWestervelt', 'Pressure', '
    PlaneWaveRadiation'], 'physics_tags': ['actd', 'actd2']},
    ...<Interface-Feature Mapping>...'
    PressureAcousticsAsymptoticScattering': {'features': [], '
    physics_tags': ['paas']}, '
    ElectromagneticWavesBoundaryElements': {'features': [], '
    physics_tags': ['embe']}, 'WallDistance': {'features': ['Wall
    '], 'physics_tags': ['wd', 'wd2']}}
```

===

Use the example provided below to infer how to format your response and generate COMSOL code.
===

EXAMPLE 0: <Same Example as in the **ModelSpecs** One-Shot Prompt>
=== Now generate the JAVA API code to compute the target quantity for the problem below. Export the table containing the target quantity to OUTPUT_PATH/output.txt.

**PROBLEM DESCRIPTION: {{ problem_description }}**
**TARGET QUANTITY to be computed: {{ target_description }}**
**ANSWER:**

We use the prompt above for the **ModelSpecs** + PhyDoc experiment, as well as to sample the initial population in the Multi-Turn Agent experiment. In the latter case, we removed the 3rd last line of code in the One-Shot example.

---

**Plan** +Phy-Doc

You are an experienced COMSOL engineer. You must generate the COMSOL API code in JAVA to execute the steps described in the plan below to compute the desired TARGET QUANTITY by generating COMSOL JAVA API code. The model creation line "'Model model = ModelUtil.create("Model");"' has already been generated and you should not repeat this line. All lines of code must begin with 'model.' You must not generate any 'model.sol...' solver code but should ensure that your 'model.study...' block ends with a 'model.study("std1").run();'. This will automatically create and run the default solver for the problem.
You are provided with the list of valid physics interfaces and features under each interface. You must only use the interfaces and features in these lists:
===
AVAILABLE COMSOL PHYSICS INTERFACES:

```
['BeamCrossSection', 'PorousMediaFlowRichards', '
    MoistureTransportInBuildingMaterials', 'CreepingFlow', '
    CathodicProtection'... <List of 140 Interface>...'
    LumpedBattery', 'CompressiblePotentialFlow', '
    BatteryBinaryElectrolyte', 'ColdPlasma', 'LaplaceEquation', '
    DilutedSpeciesInPorousCatalysts']
```

AVAILABLE FEATURES UNDER EACH INTERFACE:

```
{'ElectromagneticWavesBeamEnvelopes': {'features': ['
    MatchedBoundaryCondition', 'SymmetryPlane', 'Scattering', '
    TransitionBoundaryCondition', 'Impedance', 'Port', '
    FieldContinuity'], 'physics_tags': ['ewbe']}, '
    TransientPressureAcoustics': {'features': ['InteriorSoundHard
```

```
    ', 'InteriorLumpedSpeakerBoundary', '
    TransientMonopoleLineSource', 'CylindricalWaveRadiation', '
    Impedance', 'NonlinearAcousticsWestervelt', 'Pressure', '
    PlaneWaveRadiation'], 'physics_tags': ['actd', 'actd2']},
    ...<Interface-Feature Mapping>...'
    PressureAcousticsAsymptoticScattering': {'features': [], '
    physics_tags': ['paas']}, '
    ElectromagneticWavesBoundaryElements': {'features': [], '
    physics_tags': ['embe']}, 'WallDistance': {'features': ['Wall
    '], 'physics_tags': ['wd', 'wd2']}}
```

===
Now use the example provided below to infer how to format your response and generate COMSOL code.
===
EXAMPLE 0: PLAN: ...<Same as the One-Shot Example in **Plan** above>...
===
=== Now generate the JAVA API code to compute the target quantity for the problem below, by following the plan described. Export the table containing the target quantity to OUTPUT_PATH/output.txt.

**PLAN: {{ problem_description }}**
**TARGET QUANTITY to be computed: {{ target_description }}**
**ANSWER:**

## G.2  Multi-Turn Agent Prompts

The following prompt is used in the ToolLookupAgent to call tools. `tool_snippet` is populated with the descriptions of each tool. `state_info` is the execution and verifier feedback for the solution to iterate upon (left panel of Figure 5).

---

### Tool Selection

You are a COMSOL engineer. You are attempting to gather information relevant to execution feedback that you received from the COMSOL client after you executed some code. The relevant information can be queried as 'ToolCall'. Each 'ToolCall' must consist of str along with the relevant arguments, if any. A ToolCall may or may not require arguments. Identify the relevant tool calls and return your reply as a 'ToolCalls' object, which consists of a list of 'ToolCall's.
===
Here is some information on each tool

**{{ tool_snippet }}**

===
Now return the relevant ToolCallList for the following execution feedback / error message.

**FEEDBACK: {{ state_info }}**

---

### Correction Prompt

You are an engineer solving the following PROBLEM in COMSOL, by generating a solution that consists of the JAVA COMSOL API code needed to solve the problem. You have so far generated the code in CODE. On executing the lines in CODE you encountered the issue described in CURRENT EXECUTION FEEDBACK. CURRENT EXECUTION FEEDBACK is formatted as 'Line → Status: Error (if Status='Error')' where Status is 'Correct' if the line of code was able to execute and 'Error' if it raised an error. You have additionally been provided with EXECUTION HISTORY which is a record of some of your

previous code solutions and their execution results. You may use it as relevant context to understand what blocks of code work and what you've already tried.

You must return a BETTER solution by correcting lines of code that raised errors, or substituting blocks of code with other equivalent code snippets that would solve the problem. The solution must be a full contiguous block of CODE. Use the example provided below to understand how to format your CODE.

===

EXAMPLE 0:

PROBLEM:* Select 2D for the space dimension.

* Select Fluid Flow > Single-Phase Flow > Laminar Flow (spf).
* Create a Stationary Study
* Insert a geometry from file.
**Parameters**
* Name  Expression  Description
Re  100  Reynolds number
rho0  1e3 [kg/m$^3$]  Density

CODE:

```java
model.component().create("comp1", true);

model.component("comp1").geom().create("geom1", 2);

model.component("comp1").mesh().create("mesh1");

model.component("comp1").physics().create("spf", "FluidFlow", "
    geom1");

model.study().create("std1");
model.study("std1").create("stat", "Stationary");
model.study("std1").feature("stat").setSolveFor("/physics/spf",
    true);
model.study("std1").run();
model.component("comp1").geom("geom1").insertFile("fname.mph", "
    geom1");
model.component("comp1").geom("geom1").run("fin");

model.param().label("Geometrical Parameters");
model.param().create("par2");
model.param("par2").set("Re", "100");
model.param("par2").descr("Re", "Reynolds number");
model.param("par2").set("rho0", "1e3[kg/m^3]");
model.param("par2").descr("rho0", "Density");
...
```

EXECUTION HISTORY: CURRENT EXECUTION FEEDBACK:

```
model.component().create("comp1", true); -> Correct
model.component("comp1").geom().create("geom1", 2); -> Correct
model.component("comp1").mesh().create("mesh1"); -> Correct
model.component("comp1").physics().create("spf", "FluidFlow", "
    geom1"); -> Error: Exception com.comsol.util.exceptions.
    FlException: Unknown Interface
Messages:
        Unknown Interface
        - Interface: FluidFlow
...
```

The following information may be useful to you:
RELEVANT INFORMATION: -

NEW CODE: The Interface 'FluidFlow' is not a valid physics interface. LaminarFlow is a valid COMSOL physics interface. I will replace FluidFlow with LaminarFlow and return the entire code block.

```java
model.component().create("comp1", true);
model.component("comp1").geom().create("geom1", 2);
model.component("comp1").mesh().create("mesh1");
model.component("comp1").physics().create("spf", "LaminarFlow",
    "geom1");
model.study().create("std1");
model.study("std1").create("stat", "Stationary");
model.study("std1").feature("stat").setSolveFor("/physics/spf",
    true);
model.study("std1").run();
model.component("comp1").geom("geom1").insertFile("fname.mph", "
    geom1");
model.component("comp1").geom("geom1").run("fin");

model.param().label("Geometrical Parameters");
model.param().create("par2");
model.param("par2").set("Re", "100");
model.param("par2").descr("Re", "Reynolds number");
model.param("par2").set("rho0", "1e3[kg/m^3]");
model.param("par2").descr("rho0", "Density");
```

===

Here are some example errors, their causes, and example actions that should be taken to address them:

1. Error: 'Unknown feature'... Cause: The feature either does not exist, or is created under the wrong node. It's possible that a feature may be a defined under another feature of the interface, instead of under the interface directly. Eg: 'model.component("comp1").physics("int1").feature("f2")...' might raise an error because the correct pattern is 'model.component("comp1").physics("int1").feature("f1").feature("f2")...' Action: Ensure the feature actually exists and substitute it with a similar sounding feature if it doesn't, or define it under the correct node.

2. Error: 'Undefined material property 'A' required by FeatureNode F. Cause: An essential property needed by F (usually a solver/physics node) has not been defined correctly. Action: Edit the code where 'A' is defined. Try to set the property in one of the following ways instead. a) Easier Way. You can define a "userdefined" property under the appropriate feature branch of the 'physics' branch. The code in this case looks like:

```
model.component("comp1").physics("int1").feature("f1").set("A",
    "userdef");
model.component("comp1").physics("int1").feature("f1").set("A",
    "A_value");
```

You must have the first line, that sets the property to 'userdef' in this case, otherwise f1 might not be able to see A_value.

b) Harder Way. The property value is defined under the appropriate propertygroup of the material. The code should look like this:

```
model.component("comp1").material("mat1").propertyGroup("def
    ").set("density", "7200");
```

If the property is defined under another propertygroup of the material, the physics branch will sometimes not know where to look, and the code could fail silently.

3. Error: The code saves a value but it's far from the expected value, even though the code is executable. Cause: There might be an issue with the study code. You might be missing study settings or the 'study.run();' line which is essential for the default numerical solver

to run. You should also preferably not generate any 'model.sol' lines and ensure that the 'model.study..' block ends with 'model.study.run();' as this automatically chooses the default COMSOL solver for the problem and runs it. Action: Try to redefine the .study() code so it includes only the bare minimum described in 'Cause'.

4. Error: 'Feature cannot be created in dimension'. Cause: The feature is being created in a dimension inconsistent with the dimension of the problem. Action: Examine what the dimension of the goemetry is and reassess what the correct dimension of the feature should be. For example, a domain feature will typically have the same dimension as the geometry and a boundary feature will have D_geom -1.

5. Error: 'SelectionOutOfBoundsException: Illegal input vector illegal entity number.' Cause: An incorrect or non-existent entity number has been assigned. Action: Please recheck the SELECTION INFORMATION and ensure your code is exactly consistent with it.

Note, this is NOT an exhaustive list, and several other errors can occur. Read the error messages carefully, as they typically provide hints about the cause.

===

Now return the corrected code for the following problem:

**PROBLEM:** {{problem}}

**EXECUTION HISTORY:** {{history}}

**CURRENT CODE:**
```
{{code}}
```

**CURRENT EXECUTION FEEDBACK:** {{state_info}}

The following information may be useful to you:

**RELEVANT INFORMATION:** {{tool_lookup}}

**CORRECTED CODE:**

