# OpenReview forum: "FEABench: Evaluating Language Models on Real World Physics Reasoning Ability"
_NeurIPS.cc/2024/Workshop/MATH-AI — MATH-AI 24_

### Official Review · Reviewer_89Ha · 2024-09-29
**Evaluation of Language Models for Physics Problem-Solving Using FEA Software**

**Rating:** 6
**Confidence:** 4

**Review:**

This paper introduces FEABench, a novel benchmark for evaluating the ability of large language models (LLMs) to solve real-world physics and engineering problems using finite element analysis (FEA) software. The authors present a multi-faceted evaluation scheme to assess LLMs' capabilities in interacting with COMSOL Multiphysics software through its API to solve a range of physics and engineering problems.

Pros:
1. Novel and important contribution: The paper addresses a significant gap in evaluating LLMs on real-world engineering tasks, specifically in the domain of finite element analysis.
2. Dataset creation: FEABench includes 13 quantitatively verifiable problems across various physics domains, providing a comprehensive test of LLM capabilities.
3. Multi-faceted evaluation: The authors develop a range of metrics to assess different aspects of LLM performance, including code executability, model tree similarity, and physics-specific metrics.
4. Innovative approach: The authors develop an LLM agent equipped with the ability to interact with the FEA software API and iterate on solutions.
5. Experimentation: The paper evaluates multiple state-of-the-art LLMs and compares different prompting strategies.
Clear presentation: The methodology, metrics, and results are generally well-explained and illustrated with helpful figures and tables.

Cons:
1. Limited scope: The benchmark is currently restricted to problems solvable with COMSOL Multiphysics. Including other FEA software could enhance generalizability.
2. Relatively small dataset: With only 13 problems, the benchmark may not fully capture the diversity of real-world FEA tasks.
3. Lack of human baseline: The paper would benefit from comparing LLM performance to that of human engineers on the same tasks.
4. Potential bias in problem selection: The criteria for selecting problems could be more thoroughly justified to ensure a representative sample of FEA tasks.
5. Limited analysis of failure modes: While the paper presents performance metrics, a deeper analysis of why LLMs fail on certain tasks could provide valuable insights.
6. Reproducibility concerns: The reliance on proprietary software (COMSOL) may limit the reproducibility of the benchmark by other researchers.
7. Questionable real-world relevance: The paper doesn't adequately demonstrate how the ability to generate COMSOL API calls translates to actual problem-solving capabilities in engineering contexts.

General Comments:
- The paper makes a sizable contribution to the field of AI evaluation, specifically in assessing LLMs' capabilities in complex engineering tasks. The FEABench provides a  tool for measuring progress in applying LLMs to real-world physics and engineering problems.
- However, the benchmark's current limitations in scope and scale somewhat constrain its immediate impact. Expanding the number and diversity of problems, including problems from other FEA software packages, and providing a human performance baseline would significantly strengthen the work.
- The paper does highlight the current limitations of LLMs in solving complex physics problems, but it fails to provide substantive insights or clear paths forward. In its current form, the work serves more as a proof-of-concept for evaluating LLMs on FEA tasks rather than a robust, widely applicable benchmark.
- Despite these limitations, the paper represents an important step forward in evaluating LLMs on practical engineering tasks. It opens up new avenues for research in applying AI to complex scientific and engineering problems, potentially leading to significant advancements in how these tasks are approached in the future.

---

### Official Review · Reviewer_GwjY · 2024-10-06
**A novel idea but need more details, analysis and comparisons**

**Rating:** 5
**Confidence:** 5

**Review:**

Pros:

1.The paper addresses an important and practical problem of evaluating the ability of language models to simulate and solve physics problems using FEA software.
2. The benchmark dataset is well-curated and includes a range of problems from different domains, providing a comprehensive evaluation of the LMs' performance.
3. The multipronged evaluation scheme and the introduction of the LM agent with API interaction are innovative and provide valuable insights into the LMs' capabilities and limitations.

Cons:

1. Could you provide more details on the selection criteria and generation procedure for the benchmark dataset? How did you ensure that the problems are representative of real-world physics and engineering problems?
2. What are the potential limitations and sources of error in the evaluation scheme? How did you handle cases where the LM solutions were not executable or did not export valid target values?
3. The paper lacks a detailed analysis of the empirical findings and performance of the LMs and LM agents on the benchmark problems. The authors only provide summary statistics and do not delve into the specific challenges faced by the models or the reasons for their performance.
4. The paper does not provide a comparison with existing multi-agent methods or benchmarks in the field. It would be valuable to understand how the proposed benchmark and evaluation scheme compare to other approaches in evaluating the performance of LMs in physics reasoning tasks.
5. Can you discuss the potential biases or limitations of the benchmark? Are there any specific types of problems or scenarios that may be more challenging for LMs? How do you plan to address these limitations in future work?

---

### Official Review · Reviewer_QSaz · 2024-10-08
**Review of "FEABench: Evaluating Language Models on Real World Physics Reasoning Ability"**

**Rating:** 7
**Confidence:** 3

**Review:**

# Summary
"FEABench: Evaluating Language Models on Real World Physics Reasoning Ability" presents an innovative approach to assessing the capabilities of Large Language Models (LLMs) in solving real-world physics problems using the finite element method, a sophisticated technique often taught at the graduate level in math and physics courses. The paper makes several significant contributions to the field:
1. Development of a New Benchmark: The authors introduce FEABench, a benchmark consisting of 13 quantitatively verifiable problems derived from tutorials in the COMSOL Multiphysics⃝R Application Gallery. This benchmark is specifically designed to test the application of LLMs to complex physics problems that require substantial computational and analytical skills.
2. Multi-score Evaluation Strategy: To systematically measure the performance of LLMs on this benchmark, the paper proposes a multi-score evaluation strategy. This strategy employs a series of metrics to provide a comprehensive assessment of the models' capabilities.
3. Complexity Analysis through Prompt Engineering: The paper also explores the complexity of the dataset by employing prompt engineering techniques. The findings indicate that state-of-the-art LLMs still struggle to perform well on this challenging dataset, highlighting the need for further advancements in the field.

# Evaluation
The paper is well-organized and introduces a novel problem area in the application of LLMs to the Finite Element Method (FEM). The contributions are clearly defined and address a real gap in the current evaluation strategies for LLMs in the domain of physics.
However, the paper could be strengthened by providing a deeper connection with the broader field of numerical analysis using LLMs. A more detailed discussion on how these methods integrate with existing numerical techniques could enhance the relevance and applicability of the research.

# Recommendation
Overall, the paper is of high quality and addresses an innovative and challenging application of LLMs. The introduction of FEABench provides a valuable resource for the community, and the multi-score evaluation strategy offers a nuanced approach to assessing model performance.
Given the novelty of the problem addressed and the contributions made, I recommend the acceptance of this paper. However, I would encourage the authors to consider expanding the discussion on the integration of LLMs with traditional numerical analysis techniques to provide a more comprehensive overview of the field.

---

### Decision · Program_Chairs · 2024-10-08

Accept